# Nonlinear genomic selection index accelerates multi-trait crop improvement

J. Jesús Cerón-Rojas[1], Osval A. Montesinos-López[2], Abelardo Montesinos-López[3], Paolo Vitale[4], Paulino Pérez-Rodríguez [1], Samuel B. Fernandes[5], Rodomiro Ortiz [6] ✉ & José Crossa [1] ✉

Linear phenotypic and genomic selection indices assume additivity and linearity, limiting their ability to exploit nonlinear trait relationships. Here, we introduce the Quadratic Genomic Selection Index (QGSI), a genomic extension of the quadratic phenotypic selection index (QPSI) that integrates genomic estimated breeding values (GEBVs) within a unified quadratic framework. QGSI combines additive, squared, and cross-product terms of GEBVs, enabling phenotype-free, rapid-cycle multi-trait selection while capturing genome-wide nonlinear relationships. We evaluate QGSI using two genomic prediction strategies: (i) a maximum-likelihood additive genomic model, and (ii) a nonlinear multi-trait Gaussian kernel model that accommodates epistatic signals. Using 10 simulated maize selection cycles and two real maize and five wheat real datasets, QGSI achieves the highest selection response and the lowest prediction error variance relative to linear and quadratic phenotypic and genomic indices. Thus, combining nonlinear genomic prediction with quadratic selection indices provides a general strategy for accelerating multi-trait crop improvement.

Selection indices—linear or quadratic—have a central role in plant and animal breeding. They provide a formal framework to combine multiple traits into a single criterion for ranking individuals. The ratio indices[1] and the Smith–Hazel linear phenotypic selection index (LPSI)[2,3] offer a mathematically rigorous approach for optimizing multi-trait improvement. However, these indices rely on assumptions of linearity and additivity assumptions that are violated in biological systems where traits interact nonlinearly.

Genomic selection[4] transformed index-based breeding by enabling genome-wide, marker-prediction of breeding values. This led to the development of the linear genomic selection index (LGSI)[5], which accelerates genetic gain by shortening breeding cycles[6]. However, LGSI preserves the linear and additive structure of LPSI. Therefore, it cannot capture non-additive genetic effects, gene–gene interactions, trait–trait interactions, or intermediate trait optima[7]. The

quadratic phenotypic selection index (QPSI)[8] was introduced to overcome those limitations. This index incorporates squared and cross-product terms of traits, allowing the modeling of nonlinear relationships and stabilizing or disruptive selection patterns.

Here, we introduce the quadratic genomic selection index (QGSI), which extends QPSI by embedding genomic estimated breeding values (GEBVs)[9,10] within a quadratic selection index framework. QGSI retains the desirable statistical properties of both QPSI and LGSI while enabling the modeling of nonlinear trait contributions and genome-wide interactions through quadratic and cross-product terms of GEBVs. This index enables phenotype-free and rapid-cycle multi-trait selection[11,12]. In addition, we derive the main parameters of QPSI and QGSI: the vectors and matrices of index coefficients, mean squared prediction error (MSPE), squared correlation between the quadratic net genetic merit and its predictors, selection response, and expected

[1]Colegio de Postgraduados, Montecillos, Edo. de México, México. [2]Facultad de Telemática, Universidad de Colima, Colima, Colima, México. [3]Departamento de Matemáticas, Centro Universitario de Ciencias Exactas e Ingenierías (CUCEI), Universidad de Guadalajara, Guadalajara, Jalisco, México. [4]International Maize and Wheat Improvement Center (CIMMYT), Carretera México-Veracruz, Texcoco, Edo. de México, México. [5]University of Arkansas, Fayetteville, USA. [6]Swedish University of Agricultural Sciences (SLU) Växtförädling, Lomma, Sweden. ✉e-mail: rodomiro.ortiz@slu.se; j.crossa@outlook.com

**Table 1 | Ten simulated maize selection cycles estimated selection response (R), square correlation (SCor), and square root of the mean square prediction error (SR-MSPE)**

| Cycle | Quadratic selection index | | | | | | Linear selection index | | | | | |
|---|---|---|---|---|---|---|---|---|---|---|---|---|
| | Phenotypic | | | Genomic | | | Phenotypic | | | Genomic | | |
| | R | SCor | SR-MSPE | R | SCor | SR-MSPE | R | SCor | SR-MSPE | R | SCor | SR-MSPE |
| 1 | 163.4 | 0.844 | 40.0 | 153.5 | 0.745 | 4.0 | 19.1 | 0.942 | 2.7 | 18.4 | 0.874 | 4.0 |
| 2 | 145.2 | 0.823 | 38.3 | 158.0 | 0.730 | 4.0 | 17.1 | 0.928 | 2.7 | 18.0 | 0.866 | 4.0 |
| 3 | 122.9 | 0.798 | 35.2 | 149.9 | 0.670 | 4.6 | 15.7 | 0.916 | 2.7 | 17.8 | 0.832 | 4.6 |
| 4 | 130.2 | 0.801 | 36.9 | 171.5 | 0.791 | 3.6 | 15.9 | 0.917 | 2.7 | 19.2 | 0.900 | 3.6 |
| 5 | 121.6 | 0.795 | 35.2 | 182.7 | 0.796 | 3.6 | 15.2 | 0.911 | 2.7 | 19.5 | 0.907 | 3.6 |
| 6 | 100.9 | 0.761 | 32.2 | 157.3 | 0.724 | 4.3 | 13.7 | 0.893 | 2.7 | 17.5 | 0.842 | 4.3 |
| 7 | 87.9 | 0.750 | 28.9 | 144.1 | 0.695 | 4.0 | 13.2 | 0.890 | 2.6 | 17.9 | 0.866 | 4.0 |
| 8 | 55.3 | 0.903 | 10.3 | 82.8 | 0.997 | 1.4 | 8.8 | 0.942 | 1.2 | 9.6 | 0.998 | 1.4 |
| 9 | 46.7 | 0.888 | 9.5 | 78.2 | 0.998 | 1.2 | 7.3 | 0.916 | 1.3 | 9.1 | 0.996 | 1.2 |
| 10 | 42.6 | 0.882 | 8.9 | 80.7 | 0.996 | 1.3 | 7.3 | 0.920 | 1.2 | 9.5 | 0.995 | 1.3 |
| Average | 101.7 | 0.825 | 27.5 | 135.9 | 0.740 | 3.2 | 13.3 | 0.917 | 2.3 | 15.6 | 0.908 | 3.2 |

Maximum likelihood estimation. Results are for four traits using quadratic (quadratic (QPSI) and linear (LPSI) phenotypic selection index and quadratic (QGSI) and linear (LGSI) genomic selection index) using ML parameters estimation. Selection intensity was 10% ($k = 1.755$).

genetic gain per trait (Supplementary Methods 1 and 2). We evaluate QGSI using two estimation strategies: maximum likelihood (ML)[7] and Bayesian Gaussian kernel methods[13] (Supplementary Method 3). We assess empirical performance across 10 simulated maize selection cycles, two real maize datasets, and five real wheat datasets (Supplementary Method 4). Our results show that, by combining nonlinear genomic prediction with quadratic index theory, QGSI enables higher selection response and reduced prediction error variance. Recent study[13] has demonstrated that epistatic architectures and genome-wide interaction effects can induce substantial nonlinear covariation among traits. In this context, we incorporate a multivariate Gaussian kernel (reproducing kernel Hilbert space, RKHS)[13] framework to predict GEBVs for use in QGSI. This approach provides a flexible representation of nonlinear genomic relationships, including epistatic signals, while preserving the selection index structure.

## Results

First, we present the theoretical results for QPSI and QGSI (Supplementary Eqs. 1–21). We then describe the simulated maize datasets (Tables 1 and 2; Figs. 1 and 2), followed by the real maize datasets (Tables 3 and 4) and the real wheat datasets (Tables 5 and 6; Figs. 3 and 4). Finally, we present Supplementary Table 1–3 and Supplementary Fig. 1, which report the univariate and multivariate normality tests, as well as the estimated broad-sense heritability of the real maize and wheat datasets (Supplementary Table 4).

### QPSI theoretical results

Supplementary Equations 1–8 show that the theoretical quadratic net genetic merit ($H_q$) attains its best predictor under the ML estimator of the QPSI. This estimator is obtained by minimizing the MSPE with respect to the QPSI linear coefficient vector $b$ and the quadratic and cross-product coefficient matrix $B$. These derivations yield closed-form expressions for the minimized MSPE, the squared correlation between $H_q$ and its QPSI predictor, the selection response, and the expected genetic gain per trait, which are the basis for evaluating QPSI and comparing it with alternative selection indices.

### QGSI theoretical results

Supplementary Equations 9–16 describe the theoretical formulation of the QGSI and its main parameters. In the genomic context, QGSI is a predictor of $H_q$ and is derived by minimizing the MSPE with respect to the QGSI linear coefficient vector $\theta$ and the quadratic and cross-

product matrix $D$. These expressions yield the minimized MSPE, the squared correlation between $H_q$ and its QGSI predictor, the selection response, and the expected per-trait genetic gain.

### Maximum-likelihood and Gaussian-kernel parameter estimation

Supplementary Equations 17–21 detail the estimation of the phenotypic ($P$) and genotypic ($G$) covariance matrices, the QPSI linear and quadratic coefficients ($b, B$), and the additive genomic covariance matrix $\Gamma$. These equations also describe the estimation of trait GEBVs, LPSI, and LGSI heritabilities for real datasets, and the Bayesian multi-trait Gaussian kernel (RKHS) model used to obtain the nonlinear genomic estimators required for QGSI. Because heritability definitions for quadratic indices have not yet been formalized, QPSI and QGSI heritabilities were not estimated.

### Simulated maize selection response under maximum likelihood estimation

Across 10 simulated maize selection cycles, QPSI, QGSI, LPSI, and LGSI showed different abilities to predict $H_q$ and to generate selection response (Table 1). QPSI achieved a mean gain of 101.7, outperforming the LPSI (mean = 13.3). QGSI yielded the greatest overall response, with an average gain of 135.9 and a maximum of 182.7 in cycle C5. This advantage arises from QGSI's capacity to exploit nonlinear and interaction-driven genetic variation through squared and cross-product terms. The LGSI mean (15.6) was consistent with the limitations of an additive linear genomic model. Across all cycles, QGSI systematically outperformed LPSI and LGSI. These results align with theoretical expectations that quadratic indices extract non-additive information unavailable to linear indices. Figure 1 illustrates these differences clearly: the QGSI selection response exceeded the QPSI, LPSI, and LGSI selection responses by 25.2%, 90.2%, and 88.5%, respectively. QPSI also surpassed both linear indices by more than 80%.

### Squared correlation (predictive accuracy) between the indices and $H_q$

LPSI and QPSI (Table 1) mean accuracy were $r^2 = 0.917$ and $r^2 = 0.825$, respectively. QGSI attained a mean accuracy of $r^2 = 0.740$, striking a balance between nonlinear modeling and genomic information and maintained relatively uniform trait-specific correlations across cycles, benefiting from its ability to absorb interaction-driven signals. The LGSI mean accuracy $r^2 = 0.908$ was higher than QGSI. The QGSI

**Table 2 | Ten simulated maize selection cycles estimated expected genetic gain per trait for four traits (T1, T2, T3, T4) using phenotypic and genomic linear and quadratic indices**

| Cycle | Linear phenotypic selection index | | | | Linear genomic selection index | | | |
|---|---|---|---|---|---|---|---|---|
| | T1 | T2 | T3 | T4 | T1 | T2 | T3 | T4 |
| 1 | 10.4 | −5.5 | 3.8 | 2.0 | 8.6 | −4.7 | 3.3 | 1.8 |
| 2 | 10.1 | −4.4 | 3.7 | 2.0 | 8.8 | −3.6 | 3.4 | 2.2 |
| 3 | 9.9 | −4.1 | 3.3 | 1.7 | 8.3 | −4.0 | 2.9 | 2.6 |
| 4 | 10.9 | −4.3 | 2.6 | 1.4 | 9.7 | −4.6 | 3.1 | 1.8 |
| 5 | 10.6 | −3.5 | 3.0 | 1.5 | 9.8 | −4.5 | 3.3 | 1.9 |
| 6 | 10.0 | −3.5 | 2.5 | 1.4 | 9.0 | −3.2 | 3.2 | 2.1 |
| 7 | 5.0 | −2.0 | 1.7 | 1.4 | 7.9 | −4.7 | 2.9 | 2.4 |
| 8 | 5.6 | −1.6 | 1.2 | 1.6 | 6.5 | −0.8 | 0.8 | 1.5 |
| 9 | 5.2 | −1.5 | 0.8 | 1.4 | 6.5 | −0.5 | 0.9 | 1.2 |
| 10 | 4.9 | −1.6 | 1.3 | 1.0 | 7.1 | −0.5 | 1.0 | 0.9 |
| Average | 8.3 | −3.2 | 2.4 | 1.5 | 8.2 | −3.1 | 2.5 | 1.8 |

Maximum likelihood estimation. Results were obtained with quadratic (quadratic (QPSI) and linear (LPSI) phenotypic selection index and quadratic (QGSI) and linear (LGSI) genomic selection index) using ML parameters estimation. Selection intensity was of 10% ($k = 1.755$).

stability explains why QGSI achieved higher realized selection responses even when LGSI exhibited similar or marginally higher accuracy estimates. It is important to note that the Pearson squared correlation is inherently linear and therefore does not perfectly capture the accuracy of indices predicting a nonlinear genetic merit.

**Mean squared prediction error (MSPE)**
Despite theoretical expectations of zero asymptotic MSPE (Table 1) for LGSI and QGSI under complete genomic information, QGSI consistently achieved lower empirical MSPE than QPSI across all cycles. This result is consistent with the analytical expressions presented in Supplementary Eqs. 5 and 13. QPSI produced the highest MSPE values. LPSI showed moderate MSPE values, but only describes the linear prediction part, contrary to QGSI, which explains linear and quadratic prediction. These results highlight the importance of incorporating nonlinear genomic components and reducing prediction error.

**Expected genetic gains per trait**
Table 2 summarizes the expected genetic gains for four traits (T1, T2, T3, T4) across 10 simulated cycles using linear (LPSI, LGSI) and quadratic (QPSI, QGSI) indices under ML estimation with 10% selection intensity ($k = 1.755$). Average trait-wise gains were highly consistent between LPSI and LGSI, indicating that LGSI captured the major additive effects of the GEBVs like LPSI for predicting linear net merit. Traits with positive desired directions (T1, T3, T4) showed moderate to high gains, while T2 consistently exhibited negative correlated responses due to the imposed economic weights and covariance structure.

Figure 2 displays the average expected genetic gain across traits, highlighting the parallel behavior between phenotypic and genomic linear indices and reaffirming the robust estimation of trait-specific gains across cycles.

**Maximum likelihood estimated selection response (real maize datasets)**
Table 3 shows that for both real datasets (JMpop1 DTMA Mexico and JMpop1 DTMA Zimbabwe, hereafter JDMexico and JDZimbabwe, respectively), and across genomic selection cycles 1 and 2, QGSI outperformed LGSI in terms of selection response. The limited improvement of LGSI reflects the strictly additive nature of this index. On average, QGSI delivered 89% (JDMexico) and 80% (JDZimbabwe)

greater gains than LGSI, demonstrating a clear advantage for quadratic genomic modeling when improving correlated traits.

QPSI also outperformed LPSI in Cycle 0 (Table 3). That is, even in the absence of genomic data, QPSI is more efficient for exploiting nonlinear trait relationships. In JDMexico and JDZimbabwe, QPSI's selection response exceeded that of LPSI by 90% and 87%, respectively. These results support a phased breeding strategy: (1) Use QPSI in early cycles, and (2) Transition to QGSI once genomic predictions become feasible. These patterns are illustrated in Supplementary Method 4 for JDMexico across cycles C0–C2, and are summarized for both datasets in Tables 3 and 4 for ML and Gaussian kernel.

**Gaussian kernel estimated selection response**
When $H_q$ was predicted using the multi-trait Gaussian kernel, QGSI achieved the highest and most stable selection responses across datasets. Average QGSI gains reached 68.0 in JDMexico and 93.3 in JDZimbabwe (Table 3). In contrast, when the same Gaussian kernel predictions were used within LGSI, average gains were only 8.4 (JDMexico) and 4.6 (JDZimbabwe). This striking contrast indicates that the benefit of the Gaussian kernel is maximized when combined with a quadratic index, which can exploit non-additive signals, local epistatic patterns, and complex multi-trait covariance structures. When these nonlinear predictions are forced into LGSI, the advantage disappears.

**Maximum likelihood estimated expected genetic gain per trait (real maize datasets)**
Table 4 presents expected genetic gains for GY, PH, EHT, and AD under 10% selection intensity ($k = 1.755$). LPSI and QPSI at Cycle 0 produced larger per-trait gains for EHT and PH relative to GY and AD. Across genomic cycles, QGSI and LGSI produced, in Cycles 1 and 2, patterns similar to those of QPSI and LPSI in Cycle 0, with moderate gains for GY but larger responses for height-related traits. In JDZimbabwe, genomic indices yielded more consistent and agronomically favorable gains for PH and AD in Cycles 1 and 2 while reducing the large negative correlated responses seen under phenotypic selection. In JDMexico, GY gains at Cycle 0 were relatively higher. Still, EHT and PH gains were smaller than in Zimbabwe, underscoring the difficulty of achieving balanced multi-trait improvement when relying solely on linear or phenotypic information.

**Gaussian kernel estimated expected genetic gain per trait**
Contrary to advantage of the Gaussian kernel for selection response (Table 3), its estimates of expected genetic gain per trait were surprisingly similar to those obtained under ML (Table 4). This pattern occurs because, in this case, QGSI and LGSI give the same results (Supplementary Eq. 16). These results indicate that the value of the Gaussian kernel emerges in the context of quadratic selection indices, not in linear ones. In both maize datasets, Gaussian kernel-based QGSI and LGSI produced: (1) Higher and more stable gains for GY, (2) Moderated changes for PH and EHT, reducing extreme negative correlated responses, and (3) Minimal fluctuations for AD. These results show that QGSI combined with the Gaussian kernel yield more multi-trait improvements than phenotypic indices or ML-based genomic indices.

**Correlation between the QPSI and $H_q$**
The correlation between the QPSI and $H_q$ could be evaluated only in phenotypic cycles. Across both datasets, QPSI and LPSI produced squared correlations comparable to those observed in the simulated maize dataset (Table 1). QPSI achieved correlations between 0.39 and 0.50, while LPSI ranged from 0.55 to 0.74 (Table 3). The larger improvements under LPSI occurred in traits with moderate heritability and known interaction effects, such as those present in JDZimbabwe.

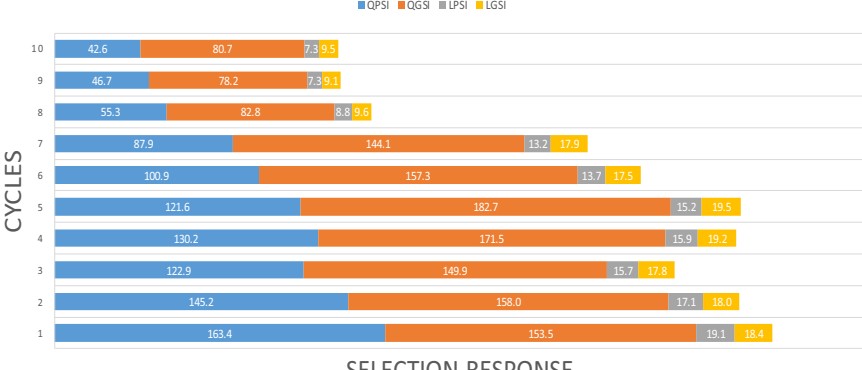

**Fig. 1 | Ten similated maize index selection response.** QPSI (quadratic phenotypic), QGSI (quadratic genomic); LPSI (linear phenotypic), and LGSI (linear genomic) selection index.

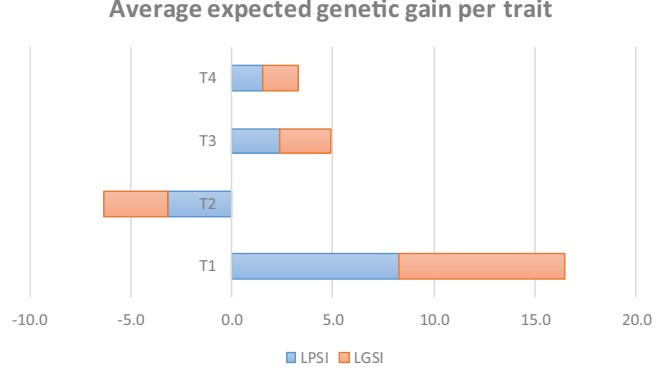

**Fig. 2 | Average genetic gain per trait.** LPSI (linear phenotypic) and LGSI (linear genomic) selection index for four traits (T1, T2, T3, T4) in 10 maize simulated selection cycles.

### Mean squared prediction error (MSPE)

In both the simulated (Table 1) and real (Table 3) datasets, during phenotypic cycles, LPSI exhibited lower MSPE than QPSI. This was particularly evident in traits subject to strong environmental fluctuations, where QPSI led to greater variation and larger error across cycles. This suggests that LPSI is more reliable when environmental variance dominates, which is the case in advanced testing stages.

### Maximum likelihood estimated selection response (real wheat datasets)

Table 5 summarizes the selection response, squared correlation, and prediction error metrics for five wheat environments evaluated using LPSI, LGSI, QPSI, and QGSI under ML and Gaussian kernel estimation. QPSI produced moderate-to-high selection responses across environments ($R = 10$–$45$), but with relatively high SR-MSPE values and only intermediate squared correlations ($SCor = 0.468$–$0.882$). By contrast, LGSI ($R = 0.7$–$4.1$) and QGSI ($R = 6.5$–$27.6$) yielded much smaller gains, reflecting the limited capacity of LGSI to capture trait complexity in wheat.

Across CENEB-BED2IR, CIANO-BED-5IR and CENEB-FLATDRIP environments, QPSI consistently delivered strong selection responses ($R = 30.4$–$45.4$) accompanied by high predictive alignment ($SCor = 0.468$–$0.882$). In contrast, LPSI selection responses remained low ($R = 1.2$–$3.2$), although exhibited lower SR-MSPE values (0.2–2.0). This pattern replicates the behavior observed in the simulated maize

datasets and is consistent with the theoretical properties shown in Supplementary Eq. 5: quadratic indices capture more signal but propagate more prediction variability. The two SAWYT environments showed similar tendencies: QPSI produced higher responses ($R = 10$–$32$) than LPSI ($R = 2.2$–$4.4$), but LPSI again obtained higher squared correlations ($SCor = 0.39$–$0.73$).

For the first top three environments individually, QGSI achieved selection responses between 6.5 and 9.0–capturing only about 14–26% of the gain observed for QPSI. LGSI produced the lowest responses ($R = 0.8$–$1.2$). For the two SAWYT environments, QGSI achieved much higher responses ($R = 11$–$28$), coming closer to the corresponding phenotypic QPSI values. When the three main environments were combined, both QPSI ($R = 29.3$; $SCor = 0.526$) and QGSI ($R = 9.0$) outperformed their linear counterparts–LPSI ($R = 1.5$; $SCor = 0.20$) and LGSI ($R = 0.70$). Yet, QPSI again showed the highest SR-MSPE (15.9), consistent with greater sensitivity to noise.

### Gaussian kernel estimated selection response

The Gaussian kernel multi-trait increased the selection response for the QGSI, while improvements in LGSI remained modest (Table 5). Environments such as CENEB-FLATDRIP and SAWYT-27 displayed high QGSI responses ($R = 73.5$ and $61.2$, respectively). Although Gaussian kernel improved LGSI relative to ML, the gains were consistently far lower than those obtained with QGSI. Across all environments combined, QGSI produced a selection response of 38.8; more than tripling the ML-based QGSI response ($R = 9.0$) and exceeding all linear indices. These consistent improvements across a broad range of wheat environments demonstrate that nonlinear genomic modeling substantially enhances the performance of quadratic indices.

Overall, the wheat results mirror the maize findings: quadratic indices deliver larger gains than linear indices, and the combination of QGSI + Gaussian kernel is the most effective strategy for capturing additive and non-additive variance. Although phenotypic indices occasionally outperform genomic ones in absolute response, genomic indices are essential for rapid-cycle breeding and for prediction in untested or future environments.

Figure 3 illustrates the estimated selection response ($R$) across six wheat environments: CENEB-BED2IR, CIANO-BED-5IR, CENEB-FLATDRIP, the combined dataset, SAWYT-27, and SAWYT-28. Figure 3 reinforce the patterns observed in Table 5: (1) Gaussian quadratic genomic indices (G-QGSI) consistently produced the highest selection responses across all environments, (2) QPSI ranked second in most environments, followed by QGSI, indicating the strong influence of quadratic modeling even without RKHS estimation, and (3) LPSI and LGSI produced lower responses in every environment.

**Table 3 | Estimated selection response (R), square correlation (SCor), and square root of the mean square prediction error (SR-MSPE) for two (JDMexico and JDZimbabwe in cycle 0) real maize datasets and two genomic selection cycles (cycle 1 and 2)**

| Dataset | Maximum likelihood parameters estimation | | | | | | | | | Gaussian kernel estimation | |
|---|---|---|---|---|---|---|---|---|---|---|---|
| | QPSI | | | | QGSI | LPSI | | | LGSI | QGSI | LGSI |
| | Cycle | *R* | SCor | SR-MSPE | *R* | *R* | SCor | SR-MSPE | *R* | *R* | *R* |
| JDMexico | 0 | 59.7 | 0.50 | 34.1 | * | 5.9 | 0.55 | 3.0 | * | * | * |
| | 1 | | | | 35.8 | | | | 3.9 | 67.0 | 8.4 |
| | 2 | | | | 37.0 | | | | 4.1 | 69.0 | 8.3 |
| Average | | | | | 36.4 | | | | 4.0 | 68.0 | 8.4 |
| JDZimbabwe | 0 | 46.7 | 0.39 | 33.3 | * | 7.8 | 0.74 | 2.7 | * | * | * |
| | 1 | | | | 34.1 | | | | 6.9 | 95.8 | 4.6 |
| | 2 | | | | 37.1 | | | | 7.3 | 90.8 | 4.5 |
| Average | | | | | 35.6 | | | | 7.1 | 93.3 | 4.6 |

*Non genomic results for that cycle

Results were obtained using quadratic (QPSI = Quadratic phenotypic selection index and QGSI = Quadratic genomic selection index) and linear (LPSI = Linear phenotypic selection index and LGSI = Linear genomic selection index) indices for a phenotypic (Cycle 0) and two genomics selection cycles (Cycle 1 and 2) using two estimation parameters methods. Selection intensity was of 10% ($k = 1.755$).

**Table 4 | Two real maize datasets estimated index expected genetic gain per trait for four traits (GY, EHT, PH, AD) using phenotypic and genomic linear and quadratic indices**

| Index | Dataset | Cycle | Estimated expected genetic gain (ML) | | | | Estimated expected genetic gain (Gaussian) | | | |
|---|---|---|---|---|---|---|---|---|---|---|
| | | | GY | EHT | PH | AD | GY | EHT | PH | AD |
| LPSI (QPSI) | JDMexico | 0 | 1.418 | −8.622 | −3.303 | −0.064 | 1.40 | 0.16 | 4.90 | 0.01 |
| LGSI (QGSI) | | 1 | 0.828 | −0.654 | 0.446 | 0.254 | 1.33 | −1.24 | 4.00 | −0.22 |
| LGSI (QGSI) | | 2 | 0.845 | −0.742 | 0.398 | 0.247 | 1.34 | −0.31 | 4.41 | −0.15 |
| LPSI (QPSI) | JDZimbabwe | 0 | −0.004 | −17.501 | −12.518 | −1.579 | 0.12 | −9.70 | -0.47 | −1.07 |
| LGSI (QGSI) | | 1 | 0.049 | −9.813 | −8.996 | −1.038 | 0.10 | −9.08 | 0.61 | −1.16 |
| LGSI (QGSI) | | 2 | 0.049 | −10.284 | −9.394 | −1.142 | 0.12 | −8.60 | 0.63 | −1.09 |

This table summarizes estimated expected genetic gain per trait four traits (GY = Grain yield, PH = Plant height trait, EHT = Ear height trait, AD = Anthesis day) were obtained for two real maize datasets for phenotypic (at cycle 0) and genomic (at cycles 1 and 2) selection using two estimation methods: ML and Gaussian kernel. Selection intensity was of 10% ($k = 1.755$).

**Table 5 | Wheat real datasets estimated selection response (R), square correlation (SCor), and square root of mean square prediction error (SR-MSPE)**

| Environment | Maximum likelihood (ML) parameters estimation | | | | | | | | Gaussian kernel estimation | |
|---|---|---|---|---|---|---|---|---|---|---|
| | QPSI | | | QGSI | LPSI | | | LGSI | QGSI | LGSI |
| | *R* | S-Cor | SR-MSPE | R | *R* | S-Cor | SR-MSPE | *R* | *R* | *R* |
| CENEB-BED2IR | 33.5 | 0.558 | 17.0 | 8.8 | 2.6 | 0.41 | 1.8 | 1.2 | 47.9 | 5.4 |
| CIANO-BED-5IR | 30.4 | 0.468 | 18.4 | 7.5 | 3.2 | 0.44 | 2.0 | 1.2 | 39.4 | 4.9 |
| CENEB-FLATDRIP | 45.4 | 0.717 | 16.3 | 6.5 | 1.5 | 0.94 | 0.2 | 0.80 | 73.5 | 6.6 |
| All above environments | 29.3 | 0.526 | 15.9 | 9.0 | 1.2 | 0.20 | 1.3 | 0.70 | 38.8 | 3.3 |
| SAWYT-27 | 31.6 | 0.687 | 12.1 | 27.6 | 4.4 | 0.73 | 1.5 | 4.1 | 61.2 | 7.5 |
| SAWYT-28 | 10.5 | 0.882 | 2.6 | 11.0 | 2.2 | 0.39 | 1.6 | 2.2 | 19.7 | 7.6 |

Results were obtained with quadratic (QPSI = Quadratic phenotypic selection index and QGSI = Quadratic genomic selection index) and linear (LPSI = Linear phenotypic selection index and LGSI = Linear genomic selection index) indices for a phenotypic (Cycle 0) and genomics selection cycles (Cycle 0) using two estimation parameters methods: ML and Gaussian kernel. Selection intensity was of 10% ($k = 1.755$).

### Maximum likelihood estimated expected genetic gain per trait (real wheat datasets)

Table 6 reports expected genetic gains for grain yield (GY), heading date (HD), and plant height (PH) at a selection intensity of 10% ($k = 1.755$). QPSI and LPSI yield identical expected gains, as do QGSI and LGSI. Phenotypic indices produced moderate positive gains in GY, modest changes in HD, and often large negative or inconsistent shifts in PH. LGSI and QGSI tended to reduce the magnitude of these extreme correlated responses, but gains in GY remained small.

For the three top environments, phenotypic indices produced higher GY gains (0.18–0.37; combined = 0.21) than genomic indices (−0.01 to 0.10; combined = 0.02). In the two SAWYT environments, however, both phenotypic and genomic indices delivered larger gains in GY (LPSI = 0.56–0.61; LGSI = 0.22–0.42), likely reflecting the strong genotype × environment interactions typical of advanced yield trials. Across those three environments and their combined set, phenotypic indices generally yielded desirable reductions in HD (e.g., −2.50; −3.90) and moderate-to-strong reductions in PH (e.g., −7.32; −6.13),

**Table 6 | Wheat traits estimated expected genetic gain per trait with two estimation parameter methods**

| Environments | Maximum likelihood (ML) parameters estimation | | | | | | Gaussian kernel estimation | | |
|---|---|---|---|---|---|---|---|---|---|
| | LPSI(QPSI) | | | LGSI(QGSI) | | | LGSI(QGSI) | | |
| | GY | HD | PH | GY | HD | PH | GY | HD | PH |
| CENEB-BED2IR | 0.37 | −2.50 | −4.05 | 0.19 | −0.32 | −0.36 | 0.48 | −0.96 | −1.33 |
| CIANO-BED-5IR | 0.36 | 0.26 | −7.32 | 0.10 | −0.05 | −2.78 | 0.46 | 0.37 | −1.35 |
| CENEB-FLATDRIP | 0.18 | −3.90 | 8.72 | −0.01 | −0.57 | −0.58 | 0.60 | −0.34 | −1.50 |
| All above environments | 0.21 | −2.18 | -6.13 | 0.02 | -0.22 | −0.31 | 0.28 | -0.65 | −1.01 |
| SAWYT-27 | 0.61 | 1.37 | 3.21 | 0.42 | 1.24 | 2.95 | 0.76 | 1.58 | 4.07 |
| SAWYT-28 | 0.56 | −1.51 | * | 0.22 | −1.40 | * | 0.76 | −0.85 | * |

This table presents expected genetic gains per trait (GY = Grain yield, HD = Heading date, PH = Plant height trait) for quadratic (QPSI = Quadratic phenotypic selection index and QGSI = Quadratic genomic selection index) and linear (LPSI = Linear phenotypic selection index and LGSI=Linear genomic selection index) indices for a phenotypic (Cycle 0) and genomics selection cycle (Cycle 0) using two estimation parameters methods: ML and Gaussian kernel.

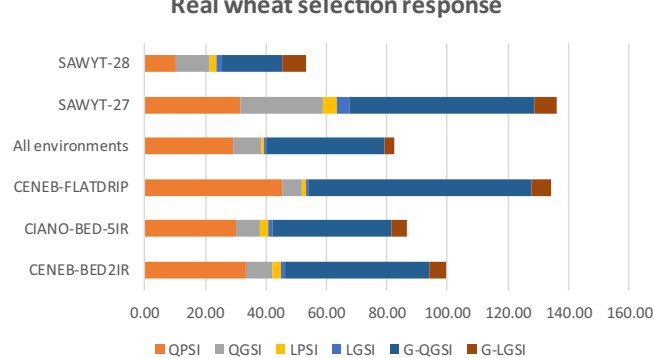

**Fig. 3 | Real wheat selection response.** QPSI (quadratic phenotypic), QGSI (quadratic genomic); LPSI (linear phenotypic), LGSI (linear genomic); G-QGSI (Gauss quadratic genomic), and G-LGSI (Gauss linear genomic) selection index across five real wheat datasets in five environments (CENEB-BED2IR, CIANO-BED-5IR, CENEB-FLATDRIP, SAWYT-27, SAWYT-28) and all bottom three environments.

supporting improved lodging resistance and plant architecture. Genomic indices also reduced HD and PH, but to a far smaller extent. In SAWYT-27, phenotypic indices produced favorable gains for GY and HD but undesirable positive shifts in PH, reflecting environment-specific correlations.

Taken together, ML results show that phenotypic indices are effective for altering height and maturity traits, but genomic indices under ML produce more conservative and less variable per-trait gains, albeit with smaller improvements.

### Gaussian kernel estimated expected genetic gain per trait

In contrast to ML, the Gaussian multitrait kernel produced more favorable and balanced expected gains across traits (Table 6). LGSI(QGSI) increased GY gains more than doubling ML-based genomic gains, while moderating negative correlated responses in HD and PH. For example: (1) CENEB-BED2IR: GY improved from 0.19 (ML) to 0.48 (RKHS), while negative shifts in HD and PH were reduced, (2) CIANO-BED-5IR: GY increased from 0.10 (ML) to 0.46 (Gaussian kernel), and undesirable decreases in PH were moderated, (3) SAWYT-27: All three traits improved under RKHS, with the largest gains observed for GY and favorable responses in HD and PH.

Across environments, the Gaussian kernel produced more stable, biologically desirable, and agronomically coherent responses than ML, highlighting the importance of modeling nonlinear genomic relationships when traits are polygenic and environments interact strongly with genotype performance. The wheat results confirm that while phenotypic indices often achieve the largest raw gains, genomic indices, under the Gaussian kernel, provide balance and stability across multiple traits, making them valuable tools for modern breeding.

Figure 4 illustrates the estimated expected genetic gain per trait across environments CENEB-BED2IR, CIANO-BED-5IR, CENEB-FLAT-DRIP, the combined dataset, SAWYT-27, and SAWYT-28. Figure 4 reinforce the patterns observed in Table 6. That is, due to the linear pattern of the estimated expected genetic gain per trait: (1) Gaussian QGSI, (2) QPSI, and (3) LPSI and LGSI produced similar results in every environment.

### Distributional normality properties in simulated data

In the simulated maize datasets, Supplementary Tables 1 and 2 show that the joint distribution of (i) the true quadratic net genetic merit with QPSI and QGSI, (ii) univariate residuals, and (iii) the multivariate distributions of traits and GEBVs all produced $p$-values above conventional rejection thresholds. Average $p$-values for traits and GEBVs ranged from 0.15 to 0.58, indicating no evidence against multivariate normality. Visual diagnostics (Supplementary Fig. 1) also confirmed that QPSI and QGSI approximated normal distributions, with only minor deviations in the tails. These results support the validity of the analytical expressions for selection response, accuracy, expected genetic gain, and index variance in the simulated framework.

### Real maize and wheat data

In the real maize datasets (Supplementary Table 3), phenotypic traits frequently deviated from multivariate normality, reflecting environmental variability and non-linear trait distributions. However, the corresponding GEBVs showed no evidence against joint multivariate normality, suggesting that genomic prediction smooths environmental noise and leads to more regular empirical distributions. In the wheat datasets (Supplementary Table 3), both traits and GEBVs conformed more closely to multivariate normality, with $p$-values generally above rejection thresholds across environments. This indicates that while raw phenotypic traits may deviate from normality—especially under stress—the genomic estimates tend to approximate normal distributions, supporting the application of parametric selection index theory.

Mild departures from normality did not materially affect the comparative performance of QPSI, QGSI, LPSI, or LGSI. Across species and environments, selection responses and predictive accuracies were stable, demonstrating that the quadratic genomic indices—particularly QGSI—are robust to modest violations of normality.

### Heritability of real maize and wheat traits

Supplementary Table 4 reports broad-sense heritability estimates for all real maize and wheat traits, as well as the heritability of the LPSI. In maize, GY showed low-to-moderate heritability (0.16–0.73), consistent with strong environmental sensitivity. EHT and PH exhibited moderate

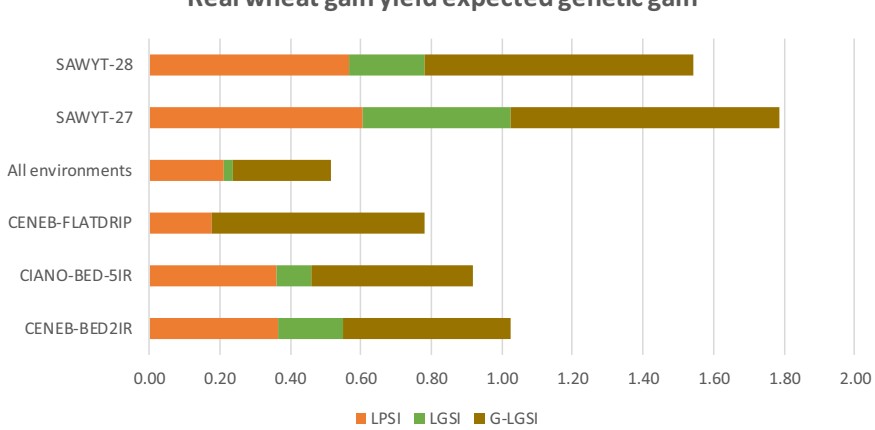

**Fig. 4 | Real wheat grain yield expected genetic gain.** LPSI (linear phenotypic), LGSI (linear genomic), and G-LGSI (Gauss linear genomic) selection index across five environments (CENEB-BED2IR, CIANO-BED-5IR, CENEB-FLATDRIP, SAWYT-27, SAWYT-28) and all bottom three environments.

heritabilities ($\approx 0.46$–$0.62$), while AD showed moderately high values ($0.51$–$0.68$). The heritability of LPSI averaged $0.64$, indicating that the phenotypic index retained a meaningful proportion of genetic variance despite heterogeneity among individual traits.

In wheat, heritability patterns were generally higher and more stable. Heading date (HD) showed very high heritability ($0.87$–$0.96$), reflecting its strong genetic control; PH had moderate-to-high values ($0.34$–$0.75$); and GY ranged from low to moderate ($0.02$–$0.73$). The LPSI heritability ($0.45$–$0.91$) demonstrated that phenotypic indices captured substantial additive genetic signal even when individual traits varied widely across environments.

These heritability patterns help explain the observed index performance: (1) Traits with higher heritability (HD, PH) produced stable and predictable responses under both linear and quadratic indices. (2) Traits with lower heritability (GY) benefited more from genomic modeling and the non-linear structure of QGSI, which captures additional variance not accessible to linear indices. (3) The strong heritability of the LPSI clarifies why phenotypic selection remains competitive in early cycles, while the extra variance captured by QGSI explains its superiority once genomic information becomes available.

## Discussion

We introduced and validated the QGSI, a nonlinear extension of QPSI and LGSI theory, where genomic data and non-additive genetic effects are simultaneously considered. Across simulated and real-world maize and wheat datasets, QGSI and QPSI outperformed LGSI and LPSI, delivering higher selection responses, stronger ranking fidelity, and more favorable genetic gains. These results reinforce the operational logic of deploying QPSI during early selection cycles when phenotypic data are still collected, and transitioning to QGSI in fully genomic pipelines.

Across both simulated datasets (Tables 1 and 2) and real maize and wheat datasets (Tables 3–6), QGSI demonstrated a unique advantage: it adapts to changing trait distributions, maintains balanced responses across multiple traits, and incorporates complex genomic patterns that LGSI cannot detect. Its capacity to account for non-additive effects, local epistatic interactions, and multi-trait genomic relationships makes QGSI a more dynamic and resilient tool than traditional LGSI.

The results from the real maize and wheat datasets (Tables 3–6) reinforce the complementary operational roles of the QPSI and QGSI. First, QPSI is the preferred index for early phenotypic cycles, where genotyping may still be limited. In these stages, QPSI delivered higher genetic gains, capturing non-linear relationships among traits and exploiting phenotypic covariance structures without requiring genomic information. Second, QGSI becomes the index of choice in later genomic cycles, where it outperforms LGSI in terms of selection response, prediction stability, and multi-trait coherence. This division of labor mirrors the trajectory of modern breeding programs, where early phenotypic selection transitions into rapid genomic cycles.

The combined empirical evidence clearly indicates that QPSI and QGSI represent a methodological advancement over their linear counterparts. While LPSI and LGSI offer stability under additive trait architectures, their predictive performance declines when genetic architectures involve non-linearities, epistasis, or strong trait interactions. In contrast, QPSI and QGSI extend the LPSI and LGSI, respectively, by incorporating squared terms and cross-product interactions, enabling them to exploit additional sources of genetic variance that linear indices cannot access.

Among these, QGSI stands out as the most powerful and adaptive tool. By integrating dense genome-wide marker data into a quadratic index formulation, QGSI identifies subtle multi-trait genomic patterns, captures non-additive genetic components, and remains robust to shifts in population structure and trait distributions across selection cycles. Unlike traditional indices with fixed weights, QGSI updates its coefficients in response to evolving data layers, making it inherently adaptive to dynamic breeding populations. From a practical breeding standpoint, the workflow distinctions are: (1) QPSI is most effective during early phenotypic stages, allowing breeders to maximize gains before genomic information becomes available or affordable, and (2) QGSI is the optimal tool in fully genomic pipelines, where prediction accuracy, rapid cycle turnover, and the integration of multiple traits are priorities. Its compatibility with Gaussian kernel estimation further amplifies its superiority in complex breeding scenarios.

The consistent of QGSI across both simulated and real datasets provides compelling justification for its adoption as a next-generation selection tool. In breeding programs involving multiple correlated traits, evolving genetic architectures, and potential non-additive effects, QGSI is poised to deliver greater and more sustained genetic gains than any linear index currently available. QPSI and QGSI are theoretical equivalents and confirm the robustness of the quadratic index framework. Computational demand increases with the number of traits, as quadratic indices require estimation of squared and interaction terms that may become unstable in small datasets. The closed-form derivations assume multivariate normality of breeding values—an assumption supported here by One-side (Shapiro-Wilk (SW), Henze-Wagner, Henze-Zirkler, and Royston) and two-side (Mardia) statistical tests[14–16] and consistent with genomic selection theory under the infinitesimal model[17–19]—but potentially violated in real populations with strong structure or selection history. Noise in

genotype information, limited sample sizes, or collinearity among traits may further complicate coefficient estimation.

Several promising extensions of QGSI could enhance its utility. Bayesian regularization[20] or kernelized regression methods[21,22] could improve estimation of quadratic weights in high-dimensional settings, especially when the number of traits or SNPs is large. Deep learning models[23] could automatically discover nonlinearities and trait–trait interactions beyond the scope of explicit cross-product terms. The integration of environmental covariates into genomic prediction using reaction norms[24], Gaussian kernels[13], or machine learning approaches[23–25] could enable G×E-informed QGSI for cross-environment predictions. QGSI may also be embedded in multi-layered prediction frameworks combining pedigree, markers, remote sensing data, and soil or climate information[26,27]. Incorporating dynamic economic weights or selection constraints within an index optimization framework[6] could further align selection decisions with breeding goals. Lastly, the development of open-source tools to implement QGSI would facilitate broader adoption and reproducibility across breeding programs[28–30].

Finally, the proposed QGSI framework directly addresses the growing recognition that epistasis and higher-order genetic interactions are fundamental components of complex trait architecture. Contemporary evidence from eco-evo-devo theory, omnigenic models, and multilayer interactome analyses shows that phenotypes emerge from dense networks of interacting loci whose effects are context-dependent and often nonlinear. Linear genomic selection indices can capture epistatic variance only indirectly through additive projections, whereas quadratic indices explicitly accommodate interaction-driven contributions via squared and cross-product terms. When combined with nonlinear genomic prediction methods—such as Gaussian kernel models that implicitly represent genome-wide epistasis—QGSI provides a biologically grounded and statistically coherent strategy for exploiting genetic networks without requiring explicit specification of interaction terms. This integration aligns with recent theoretical and empirical advances demonstrating that accounting for epistasis improves prediction accuracy, selection response, and long-term genetic gain in complex traits[31–34]. Thus, QGSI should be viewed not merely as an extension of classical index theory, but as a network-aware genomic selection framework consistent with modern views of genetic architecture.

We conclude that QGSI represents a significant advancement over traditional linear frameworks in both conceptual depth and operational impact. By explicitly modeling non-linear trait relationships and interaction effects, QGSI enables more accurate, stable, and biologically informed selection decisions in multi-trait genomic prediction. Unlike static linear indices, QGSI adapts to changes in population structure, trait distributions, and selection pressures across cycles. It leverages genomic data not only to predict additive effects, but also to uncover complex genetic architectures, including epistasis and pleiotropy[30]. From an operational standpoint, QPSI is most effective in early selection cycles when phenotypic data are available and genomic resources are limited. In contrast, QGSI is ideally suited for high-throughput genomic selection pipelines, where rapid cycle turnover and multi-trait optimization are essential. The empirical evidence presented here—across both controlled simulations and diverse real-world maize and wheat datasets—demonstrates that QGSI consistently delivers higher and more stable genetic gains than its linear analogues. As breeding programs increasingly aim to optimize multiple, sometimes antagonistic traits under variable conditions, QGSI provides a robust, adaptive, and forward-looking framework for accelerating genetic improvement.

## Methods
### Maize real dataset
We took two maize datasets from the published literature[5]. These datasets have been denoted as JMpop1 DTMA Mexico optimum environment and JMpop1 DTMA Zimbabwe optimum environment. We used these two datasets to perform phenotypic on a training population (C0), and genomic selection on two testing population: cycle 1 (C1) and cycle 2 (C2). For each data set, C0 contained genotypic (markers) data and four phenotypic traits: grain yield (GY, t/ha), plant height (PH, cm), ear height (EHT, cm), and anthesis days (AD, d), as well as three sets of markers corresponding to C0, C1, and C2. The numbers of individuals for the two datasets were 247, each one with two repetitions, and the numbers of molecular markers was 195 for C0, C1, and C2. Assuming that the breeding objective was to increase GY while decreasing PH, EHT, and AD, the vectors of economic weights in C0, C1, and C2 was $\mathbf{w}' = [\,5 \quad -0.3 \quad -0.3 \quad -1\,]$, for all four indices and datasets. The QPSI and LPSI was applied only in C0 because there were no phenotypic data after that cycle, whereas QGSI and LGSI were applied in C1 and C2. The top 10% ($k = 1.75$) was selected in all cycles of the four data sets.

### Wheat real datasets
One wheat data comprises two years of international nursery under high rainfall environments (High Rainfall Wheat Yield Trials, HRWYT), 27th and 28th, including 3 environments in Mexico, 2 at the Cd. Obregon CIMMYT experimental station (CENEB) and 1 at CIANO station. The environments included bed planting under 2 irrigations (BED2IR), bed planting under 5 irrigations (BED5IR), and flat planting under drip irrigation (FLATDRIP). The traits measured were grain yield (GY, t/h), heading date (HD, days), and plant height (PH, cm). The sample size of genotypes or individuals was of 50, each with two repetitions, whereas the number of markers was of 3111. The vector of economic weights for GY, HD, and PH was $\mathbf{w}' = [10 \quad -0.3 \quad -0.1]$.

The other real wheat data set used in this study is SAWYT-27 and SAWYT-28, which are consecutive cycles of CIMMYT's Semi-Arid Wheat Yield Trials, designed to evaluate elite spring wheat lines under drought- and heat-prone environments. Each nursery contains approximately 40–50 advanced genotypes plus international checks, tested across diverse semi-arid locations in Mexico, South Asia, West Asia, North Africa, and Sub-Saharan Africa. Field trials typically follow an alpha-lattice design, with standardized management to impose moderate-to-severe water-limited conditions. Both crop seasons included GY, HD, and PH. Their multi-environment structure enables evaluation of yield stability, stress adaptation, and the transferability of predictive models across years. Also, this dataset included two locations at CENEB and one at CIANO with different planting systems and under drought and heat management.

### Simulated datasets
The datasets were simulated with QU-GENE[5] software using 2806 molecular markers and 315 quantitative trait loci (QTLs) for eleven phenotypic selection cycles (C0 to C10), each with four traits (T1, T2, T3, and T4), 500 genotypes, and four replicates for each genotype. The authors distributed the markers uniformly and the QTLs randomly across 10 chromosomes to simulate maize (*Zea mays* L.) populations. A different number of QTLs affected each of the four traits: 300, 100, 60, and 40, respectively. The common QTLs affecting the traits generated genotypic correlations of −0.5, 0.4, 0.3, −0.3, −0.2, and 0.1 between T1 and T2, T1 and T3, T1 and T4, T2 and T3, T2 and T4, T3 and T4, respectively. The economic weights used in the selection process for T1, T2, T3 and T4 were 1, −1, 1, and 1, respectively. We used 10 phenotypic and genomic selection cycles (C1-C10) with a proportion of selection of 10% ($k = 1.755$) in each cycle. We selected all four traits in each selection cycle.

### The quadratic genomic selection index estimation
In the QGSI context, the total genetic merit of an individual is modeled as a quadratic function of true breeding values, where the coefficients assigned to squared trait values and cross-product terms reflect their

relative economic importance. In this context, additive genetic values are estimated from SNP (Single Nucleotide Polymorphism) marker data using GBLUP (Genomic Best Linear Unbiased Prediction) and Bayesian prediction models, while the genomic relationship matrix ($\boldsymbol{\Phi}$, Supplementary Eqs. 9.1-9.3) is used to derive the index parameters. This generalization of the classical index framework enables breeders to construct selection indices that: (1) exploit both additive and non-additive genetic variance, (2) incorporate trait interactions, (3) operate without phenotypic records if necessary, and (4) remain flexible across changing populations and breeding cycles.

The quadratic net genetic merit of individuals (Supplementary Eq. 1) enables breeders to account for both additive contributions through t($\mathbf{w}$)$\mathbf{g}$ and interactions or nonlinearities among traits through t($\mathbf{g}$)$\mathbf{W}\mathbf{g}$. In turn, the QGSI (Supplementary Eq. 10) is an extension of the QPSI, which predicts Supplementary Eq. 1 based on both linear and quadratic combinations of breeding values. In genomic contexts, true genomic breeding values are unknown and are replaced with GEBVs, which have been calculated using GBLUP or Bayesian prediction models [5,6]. After the first selection cycle, each individual's GEBVs for trait $t$ is modeled as

$$\hat{\gamma} = M\beta_{BLUP} \tag{1}$$

where $\mathbf{M} = \mathbf{I}_t \otimes \mathbf{X}$ is the matrix of standardized marker genotypes (2-2p, 1-2p and -2p for genotypes AA, Aa, and aa, respectively; p is the frequency of allele A, and 1-p is the frequency of allele a, respectively) in the testing population, where $\mathbf{I}_t$ is an identity matrix of size $t \times t$, $\otimes$ denotes the direct product, and $\boldsymbol{\beta}_{BLUP}$ is the vector of GBLUPs predictor of all estimated marker additive effects for $t$ traits estimated in the training population, whereas matrix $\mathbf{X}$ was defined in Supplementary Eq. 9.1.

The estimated QGSI (Supplementary Eq. 10) integrates a linear and a quadratic component and is computed as:

$$\hat{\mathbf{I}}_{\mathbf{qg}} = t(\mathbf{w})\hat{\boldsymbol{\gamma}} + t(\hat{\boldsymbol{\gamma}})\mathbf{W}\hat{\boldsymbol{\gamma}} \tag{2}$$

where $\mathbf{w}$ is the vector of linear weights and t($\mathbf{w}$) its transpose; $\hat{\boldsymbol{\gamma}}$ is the vector of GEBVs and t($\hat{\boldsymbol{\gamma}}$) its transpose whereas $\mathbf{W}$ is a symmetric matrix of quadratic and cross products weights. The first term t($\mathbf{w}$)$\hat{\boldsymbol{\gamma}}$ captures the linear additive genetic merit, while the second term t($\hat{\boldsymbol{\gamma}}$)$\mathbf{W}\hat{\boldsymbol{\gamma}}$ model interactions or nonlinear trait GEBV contributions.

In Supplementary Eqs. 10-12, we derive the expected value and variance of the QGSI, which can be estimated as tr($\mathbf{W}\hat{\boldsymbol{\Gamma}}$) and t($\hat{\boldsymbol{\gamma}}$)$\hat{\boldsymbol{\Gamma}}\mathbf{w} + 2\text{tr}[\mathbf{W}\hat{\boldsymbol{\Gamma}}\mathbf{W}\hat{\boldsymbol{\Gamma}}]$, respectively, where $\hat{\boldsymbol{\Gamma}}$ an estimator of the genomic covariance matrix $\boldsymbol{\Gamma}$ (Supplementary Eqs. 9.2, 9.3, 19.1, and 19.2). These derivations provide theoretical support for the formulation and comparison of QGSI with linear indices under various scenarios of trait correlation and heritability. Similar results can be finding for the QPSI.

### R-software for testing the assumption of normality

We corroborated the residuals normality assumption using the stats R-package, whereas to corroborate the assumption of joint bivariate normality distribution of H and QPSI, and H and QGSI, we used the mvShapiroTest R-package (Supplementary Table 1). The stats R-package is part of the R-base software and includes the univariate one-side Shapiro−Wilk test through the shapiro.test() R-function, whereas the mvShapiroTest R-package contains the multivariate one-side Shapiro−Wilk test through the mshapiro.test() function. In addition, in Supplementary Table 2 and 3, we present the one-side (Shapiro-Wilk (SW), Henze-Wagner, Henze-Zirkler, and Royston) and two-side (Mardia) five test statistics[14–16] and their p-values to test join multivariate normality of the simulated and real datasets trait and GEBVs, respectively. Those test statistics were implemented using the R package MVN[14].

The $p$ − value was the decision criterion to accept or reject the null hypothesis ($H_0$), e.g.,

$H_0 : H_q$ and QPSI have joint bivariate normal distribution, vs.

$H_1 : H_q$ and QPSI does not have a joint bivariate normal distribution.

The $p$ − value is an estimate of the probability that a particular value of the statistical test result, or a result more extreme than the statistical test value observed, could have occurred by chance if $H_0$ were true. This is a measure of the credibility of $H_0$ and if this is large (e.g., $p = 0.30$) means that there is no evidence to reject $H_0$ for significance levels of 0.05, 0.10, or 0.20, for example ref. 16.

### Reporting summary
Further information on research design is available in the Nature Portfolio Reporting Summary linked to this article.

## Data availability

The simulated and real maize datasets are available in the Application of a Genomic Selection Index to Real and Simulated Data repository [https://hdl.handle.net/11529/10199]. The simulated phenotypic datasets are in a folder denoted as PSI_Phenotypes-05, whereas the genomic data are in a folder denoted as GSI_Phenotypes-05. Each of the two maize real datasets is in a folder denoted DATA_SET-3 and DATA_SET-4. The real wheat datasets can be downloaded from CIMMYT Research Data [https://doi.org/10.71682/10549385].

## Code availability

The R codes for the simulated and real maize and wheat datasets are available and can be downloaded from CIMMYT Research Data [https://doi.org/10.71682/10549385].

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

## Acknowledgements

We acknowledge the International Maize and Wheat Improvement Center (CIMMYT) for providing access to maize and wheat phenotypic and genomic datasets used in this study. We thank the breeding teams and collaborators involved in generating and curating the JMpop1 DTMA maize datasets and the international wheat trial datasets (including HRWYT and SAWYT). This work benefited from scientific discussions and institutional support from Colegio de Postgraduados, CIMMYT, the University of Colima, the University of Guadalajara, the University of Arkansas, and the Swedish University of Agricultural Sciences.

## Author contributions

J.J.C.-R. conceived the study, developed the theoretical framework for the quadratic genomic selection index (QGSI), and led the mathematical derivations. O.A.M.-L. contributed to the statistical modeling, simulation design, and co-wrote the Results and Methods sections. A.M.-L. implemented statistical tests, including normality diagnostics, and contributed to manuscript revision. P.V. curated wheat phenotypic and genomic datasets, performed trait standardization, and assisted in visualizations. P.P.-R. advised on genomic prediction modeling and interpretation of results across real and simulated datasets. S.F. contributed to maize dataset preparation, selection response calculations, and data validation. R.O. provided strategic insights into plant breeding interpretation and reviewed the manuscript for agricultural relevance and policy alignment. J.C. supervised the study, co-developed the conceptual design, oversaw statistical validation, and finalized the manuscript for submission.

## Funding

## Competing interests

The authors declare no competing interests.
