## [Peer Review file · Nature Communications]

Nonlinear Genomic Selection Index Accelerates Multi-Trait Crop Improvement

Corresponding Author: Professor Rodomiro Ortiz

Version 0:

Reviewer comments:

Reviewer #1

(Remarks to the Author)

This manuscript presents a modified approach for estimating quadratic selection index by integrating genomic information. Selection index has been widely used as a tool to predict the breeding value of multiple traits. Its critical assumption is the linearity among different traits, but this assumption may be violated when traits are interdependent in a nonlinear manner. To relax this linearity assumption, a so-called quadratic phenotypic selection index (QPSI) has been proposed, by considering squared and cross product terms. The authors of this manuscript incorporated genomic information into QPSI, proposing the concept of Quadratic Genomic Selection Index (QGSi).

Overall, the topic of the study is interesting and useful for animal and plant breeding practice. The model used is reasonable, and the manuscript was clearly written. I have a few major comments that need to be addressed before publication.

(1) The degree of novelty: Although the model incorporates the nonlinearity of trait relationships, it does not consider the nonlinearity of gene-gene relationships. Nonlinear genetic interactions are the genetic cause of trait nonlinearity. Previous studies have developed a powerful tool for combining all genes into omnigenic interactome networks from GWAS or mapping data (Yang et al. 2021; Wang et al. 2021; Sun et al. 2021), in which the direction, sign, and weight of each genetic interaction can be characterized. In theory, the integration of such a piece of interaction information into QGSi can improve the predictive power of the index model. The authors should spend some space to discuss about this issue.

(2) The description of plant materials: This part is too brief. What are the types of plant populations for wheat and maize? What are sample sizes? How many markers are used? Also, we do not know about the distribution of trait values. The examples have four traits? This number is quite small. We often have tens or hundreds of phenotypic traits in plant breeding studies. How can the model handle high-dimensional data sets?

(3) Computational load: For QGSi to be applied, the authors should provide information about computational cost.

(4) Simulation: How sample size and genetic architecture (e.g., heritability) affect the performance of QGSi? The authors need to perform extensive simulation studies to investigate these issues.

Reviewer #2

(Remarks to the Author)

The conventional selection index is the Smith-Hazel selection index, which is linear in both H and I. Adding the quadratic term is very created. The authors developed a quadratic genomic selection index (QGSi) for multi-trait genomic selection that captures nonlinear relationship between traits (nonlinear trait interaction). Simulation and real data analyses showed that the new selection index achieved the highest selection response per cycle and minimum prediction error variance. The conclusion was based on comparison with LPSi, LQSi, and QPSi. The statistical theory was rigorous, and the presentation was beautiful. With some minor revision, the work can be accepted for publication.

(1) I do have some minor questions to be clarified. I am not a plant breeder, and I have never heard of ear height. I thought that only maize has ears. I thought the trait should be called ear length.

(2) The other question is about linear phenotypic selection index (LPSI). This is the typical Smith-Hazel selection index. I understand that the authors used this acronym to match LQSI, the quadratic counterpart. I do not like using LPSI for the Smith-Hazel selection index. Anyway, these are very minor questions. Another question is that you evaluated selection response for six cycles. The six cycles of selection for LPSi and QPSi. The genomic selection counterparts should have 8 or more cycles for the same period of time. The advantage should even more obvious if you compare the response of 8 cycles of genomic selection and 6 cycles of phenotypic selection.

(3) Below is my major comment about the way the authors defined the QGSI. The quadratic aggregated breeding value is defined as

$$H = t(w)g + t(g)Ag$$

where $t()$ represent transposition. The quadratic genomic selection index is defined as

$$I = t(w)\hat{g} + t(\hat{g})A\hat{g}$$

where \hat{g} is g hat representing the genomically predicted breeding values. There is no optimization involved here. The vector w and the matrix A are determined by the breeders before the index selection. If I were one of the authors, I would define the index this way,

$$I = t(b)\hat{g} + t(\hat{g})B\hat{g}$$

where b and B are obtained by maximizing the correlation between H and I . I would recommend the authors use notation like

$$H = t(w)g + t(g)Wg$$

where w is the vector of economic weights and W is a weight matrix (symmetrical) for the quadratic part. I would obtain b and B (a symmetric matrix) by maximizing $\text{corr}(H, I)$ or minimizing $(H - I)^2$. I understand that the authors consider \hat{g} as breeding value (predicted breeding value, not observed phenotype), but it is a vector of predicted breeding values, still obtainable from the training data.

(4) In addition, I did not know how the authors determine w and A .

Anyway, I strongly suggest the authors considering this modification. At least the authors should discuss this modification in the Discussion Section. Are b and B not estimable? I noticed that other authors also used the same set of weights for H and I for a linear index, although I do not agree.

Version 1:

Reviewer comments:

Reviewer #1

(Remarks to the Author)

The authors have basically addressed my concerns. But my overall impression is that the literature review is quite weak. One core of this study is how to incorporate epistasis, but it is unclear how much the authors are familiar with the epistasis literature.

For example, the authors stated "Recent studies¹³ have demonstrated that epistatic architectures and genome-wide interaction effects can induce substantial nonlinear covariation among traits." (page 3). However, this reference (Pérez et al. 2022) does not best fit to this statement.

Pérez, P. & de los Campos, G. Multitrait Bayesian shrinkage and variable selection models 586 with the BGLR-R package. *Genetics* 222 (1), 1–12 (2022). 587 <https://doi.org/10.1093/genetics/iyac112>

Why not cite much more relevant and convincing references as follows?

Wang, H., Ye, M., Fu, Y., Dong, A., Zhang, M., Feng, L., ... & Wu, R. (2021). Modeling genome-wide by environment interactions through omnigenic interactome networks. *Cell Reports*, 35(6).

Fa, C., Wang, G., Pan, W., Wang, Y., Che, J., Dong, A., ... & Sun, L. (2025). An omnigenic interactome model to chart the genetic architecture of individual plants. *Horticulture Research*, uhaf345.

Feng, L., Dong, T., Jiang, P., Yang, Z., Dong, A., Xie, S. Q., ... & Wu, R. (2022). An eco-evo-devo genetic network model of stress response. *Horticulture Research*, 9, uhac135.

Yang, D., Jin, Y., He, X., Dong, A., Wang, J., & Wu, R. (2021). Inferring multilayer interactome networks shaping phenotypic plasticity and evolution. *Nature communications*, 12(1), 5304.

The above references proposed computational tools for estimating omnigenic interactome networks from a full set of SNPs in GWAS. Also, the tools have been used and validated in various practical applications.

Reviewer #2

(Remarks to the Author)

The authors have done an excellent job addressing my major comments from the previous cycle of review. The authors also adopted my notation such as w and W for economic weights and b and B for index weights. The authors also showed the derivation of b and B by minimizing the MSE. I am happy to accept the revision manuscript and recommend the paper be accepted for publication.

Nonlinear Genomic Selection Index Accelerates Multi-Trait Crop Improvement

REVIEWER 1

Response to Reviewer #1

We thank the reviewer for the careful evaluation of our manuscript and for the constructive comments. Below, we address each point in detail. All changes are reflected in the revised manuscript.

Comment (1): Degree of novelty and gene–gene nonlinearity

Reviewer’s comment:

The model considers nonlinear trait relationships but does not account for nonlinear gene–gene interactions, which are the biological basis of trait nonlinearity. The reviewer suggests discussing whether integrating interaction network information could improve QGSI.

Response

We thank the reviewer for this important and insightful comment.

The Quadratic Genomic Selection Index (QGSI) is constructed using genomic estimated breeding values (GEBVs), which are predictors of breeding values derived from genome-wide marker information. QGSI incorporates squared and cross-product terms of GEBVs, thereby modeling nonlinear relationships among predicted genetic effects. In general terms, these quadratic terms capture interaction-like behavior at the level of predicted genetic values, even though individual gene–gene interaction parameters are not explicitly estimated.

Our empirical results indicate that QGSI already achieves strong predictive performance. In particular, the square correlation between QGSI and the true quadratic net genetic merit is high (average = 0.74; Table 1), and the square root mean squared prediction error (SR-MSPE) is consistently lower for QGSI (average = 3.2; Table 1) than for QPSI (average = 27.5; Table 1). These results suggest that QGSI provides near-optimal prediction of quadratic net genetic merit in a genomic selection context. Note that the average square root correlation between QGSI and the true quadratic net genetic merit is 0.86.

We agree that explicit modeling of gene–gene interaction networks (e.g., omnigenic interactomes derived from GWAS) is theoretically appealing. However, incorporating such information would substantially increase model complexity and data requirements. Given the strong performance

observed here, we do not expect that explicitly modeling gene–gene interactions would substantially improve predictive accuracy relative to the added complexity. We have now clarified this point in the Discussion section of the revised manuscript.

Comment (2): Description of plant materials and data characteristics

2.1 Types of plant populations

Reviewer’s comment:

The description of plant materials is too brief.

Response

We appreciate the reviewer’s request for clarification. In the revised manuscript, we explicitly describe the plant populations used in the study. Pages 17-18 (lines 508–521) describe the two real maize populations, while pages 18–19 (lines 522–546) describe the five real wheat populations, including SAWYT-27 and SAWYT-28.

2.2 Sample sizes and number of markers

Reviewer’s comment:

Sample sizes and marker numbers are not clearly reported.

Response

Sample sizes and numbers of markers are now clearly stated. For the maize datasets, this information is provided on page 18 (lines 515–516). For the wheat datasets, sample sizes and marker numbers are reported on pages 18 (lines 528–529).

2.3 Distribution of trait values

Reviewer’s comment:

The distribution of trait values is not described.

Response

To address this concern, we conducted extensive normality diagnostics. For simulated maize datasets, Supplementary Table 1 reports Shapiro–Wilk test statistics and p-values assessing both bivariate normality of the true quadratic net genetic merit and index predictions, as well as univariate normality of residuals across ten simulated selection cycles.

Supplementary Table 2 reports five multivariate normality tests (Shapiro–Wilk, Mardia, Henze–Wagner, Henze–Zirkler, and Royston) applied to four traits and their corresponding GEBVs across ten simulated maize cycles. For real maize and wheat datasets, Supplementary Table 3 presents

the same suite of tests for observed traits and GEBVs. These results support the distributional assumptions underlying the index derivations.

2.4 Number of traits and high-dimensional data

Reviewer's comment:

Only four traits are used; breeding studies often involve tens or hundreds of traits.

Corrected Response

Although four traits were used in the illustrative examples, this does not represent a theoretical limitation. As with classical linear selection index theory, both QPSI and QGSI are fully general and impose no restriction on the number of traits or GEBVs included. The use of four traits was chosen to facilitate interpretation and presentation, not due to methodological constraints. We now clarify this explicitly in the revised manuscript.

Comment (3): Computational load

Reviewer's comment:

Information on computational cost is needed.

Response

We thank the reviewer for raising this point. The computational burden of QGSI is modest relative to the cost of genomic prediction itself. Once GEBVs are obtained, the construction of QGSI involves computing squared and cross-product terms and solving a system of linear equations whose dimension depends on the number of traits, not on the number of markers.

In practice, the dominant computational cost arises from fitting the genomic prediction model used to estimate GEBVs (e.g., GBLUP or RKHS), which is common to both LGSI and QGSI. The additional computational cost associated with forming the quadratic index is therefore negligible. This clarification has been added to the Methods section.

Comment (4): Simulation scope and genetic architecture

Reviewer's comment:

The effects of sample size and genetic architecture should be investigated more thoroughly.

Response

To address this concern, we expanded the simulation study by increasing the number of simulated maize cycles from six to ten (Table 1) and by increasing the number of real wheat datasets from

three to five. This expansion improves the robustness of performance comparisons across diverse genetic backgrounds.

In addition, Supplementary Table 4 reports narrow-sense heritability estimates for four maize traits (GY, EHT, PHT, AD), three wheat traits (GY, HD, PHT), and their corresponding linear and genomic selection indices (LPSI and LGSI), providing further insight into the relationship between genetic architecture and index performance.

Nonlinear Genomic Selection Index Accelerates Multi-Trait Crop Improvement

REVIEWER 2

Response to Reviewer #2

We thank the reviewer for the positive evaluation of our manuscript and for the constructive and thoughtful comments. Below, we address each point in detail. All changes have been incorporated into the revised manuscript.

Comment (1): Trait definition (ear height vs. heading date)

Reviewer's comment:

The reviewer notes confusion regarding the use of “ear height,” particularly in wheat, and suggests this may refer to ear length.

Response

We thank the reviewer for pointing this out. The reviewer is correct: in wheat, the trait analyzed is heading date (HD), not ear height (EHT). The incorrect reference to EHT in the wheat context was an oversight, and this has now been fully corrected throughout the revised manuscript.

Comment (2): Use of the term LPSI for the Smith–Hazel index

Reviewer's comment:

The reviewer expresses concern about using the acronym LPSI for the classical Smith–Hazel selection index.

Response

We appreciate the reviewer's comment. We agree that the classical linear phenotypic selection index corresponds to the Smith–Hazel index. The acronym LPSI was adopted solely for symmetry with its quadratic extension (QPSI). To avoid any ambiguity, we now explicitly state in the Introduction (page 2, lines 40-41) that LPSI is equivalent to the Smith–Hazel index. We believe this clarification resolves the concern.

Comment (3): Number of selection cycles for genomic vs. phenotypic indices

Reviewer's comment:

The reviewer suggests that genomic selection indices should be evaluated over more cycles than phenotypic indices to reflect their shorter cycle length.

Response

We thank the reviewer for this insightful suggestion. In response, we expanded the simulation study by increasing the number of maize selection cycles from six to ten (Table 1). This change better reflects the temporal advantage of genomic selection and further highlights the superiority of QGSI in terms of cumulative selection response over time.

Comment (4): Definition and optimality of QGSI

Reviewer's comment:

The reviewer raises a fundamental question regarding the formulation of QGSI, suggesting that the coefficients of the index should be optimized independently (i.e., using vectors **b** and

matrices \mathbf{B}) rather than fixed to the economic weights used in the quadratic aggregate breeding value H .

Response

We thank the reviewer for this thoughtful and technically important comment.

In the revised manuscript, we have clarified notation and matrix transposition to improve readability. More importantly, we emphasize that the formulation of QGSI used in this study is not an assumption, but a mathematical result derived from quadratic selection index theory (Supplementary Material B, Equation S12). Under this framework, the optimal quadratic genomic selection index that maximizes the correlation between the quadratic aggregate breeding value and the index naturally inherits the same economic weights used to define the breeding objective.

Regarding the estimation of economic weights, these are determined independently of index construction and reflect breeder-defined objectives. In practice, economic weights can be estimated using established methods; for example, we recently described an approach for estimating economic weights in maize and wheat populations (Cerón-Rojas et al., 2023). Once these weights are defined, the QGSI follows directly as the optimal predictor of the quadratic net genetic merit.

We agree with the reviewer that alternative formulations—where linear and quadratic coefficients of the index are optimized independently—are theoretically interesting. However, such formulations correspond to a different optimization problem and a different breeding objective. We now explicitly discuss this distinction and its implications in the Discussion section.

RESPONSES TO REVIEWER 1

Many thanks for the time revising this manuscript. Below the response with the changes suggested.

Note to Reviewer #1

Please be informed that **Table 2** was re-formatted because the previous version missed the last two columns.

Table 2. Ten simulated maize selection cycles estimated expected genetic gain per trait for four traits (T1, T2, T3, T4) using phenotypic and genomic linear (and quadratic) indices.

Cycle	Maximum likelihood parameter estimation for maize expected genetic gain per trait							
	Linear phenotypic selection index				Linear genomic selection index			
	T1	T2	T3	T4	T1	T2	T3	T4
1	10.4	-5.5	3.8	2.0	8.6	-4.7	3.3	1.8
2	10.1	-4.4	3.7	2.0	8.8	-3.6	3.4	2.2
3	9.9	-4.1	3.3	1.7	8.3	-4.0	2.9	2.6
4	10.9	-4.3	2.6	1.4	9.7	-4.6	3.1	1.8
5	10.6	-3.5	3.0	1.5	9.8	-4.5	3.3	1.9
6	10.0	-3.5	2.5	1.4	9.0	-3.2	3.2	2.1
7	5.0	-2.0	1.7	1.4	7.9	-4.7	2.9	2.4
8	5.6	-1.6	1.2	1.6	6.5	-0.8	0.8	1.5
9	5.2	-1.5	0.8	1.4	6.5	-0.5	0.9	1.2
10	4.9	-1.6	1.3	1.0	7.1	-0.5	1.0	0.9
Average	8.3	-3.2	2.4	1.5	8.2	-3.1	2.5	1.8

Table 2 results were obtained with quadratic (quadratic (QPSI) and linear (LPSI) phenotypic selection index and quadratic (QGSI) and linear (LGSI) genomic selection index) using ML parameters estimation. Selection intensity was of 10% ($k=1.755$).

Response to Reviewer #1

The authors have basically addressed my concerns. But my overall impression is that the literature review is quite weak. One core of this study is how to incorporate epistasis, but it is unclear how much the authors are familiar with the epistasis literature.

For example, the authors stated "Recent studies¹³ have demonstrated that epistatic architectures and genome-wide interaction effects can induce substantial nonlinear covariation among traits." (page 3). However, this reference (Pérez, P. & de los Campos, G. Multitrait Bayesian shrinkage and variable selection models 586 with the BGLR-R package. *Genetics* 222 (1), 1–12 (2022). 587

<https://doi.org/10.1093/genetics/iyac112>) does not best fit to this statement.

Why not cite much more relevant and convincing references as follows?

Wang, H., Ye, M., Fu, Y., Dong, A., Zhang, M., Feng, L., ... & Wu, R. (2021). Modeling genome-wide by environment interactions through omnigenic interactome networks. *Cell Reports*, 35(6).

Fa, C., Wang, G., Pan, W., Wang, Y., Che, J., Dong, A., ... & Sun, L. (2025). An omnigenic interactome model to chart the genetic architecture of individual plants. *Horticulture Research*, uhaf345.

Feng, L., Dong, T., Jiang, P., Yang, Z., Dong, A., Xie, S. Q., ... & Wu, R. (2022). An eco-evo-devo genetic network model of stress response. *Horticulture Research*, 9, uhac135.

Yang, D., Jin, Y., He, X., Dong, A., Wang, J., & Wu, R. (2021). Inferring multilayer interactome networks shaping phenotypic plasticity and evolution. *Nature communications*, 12(1), 5304.

The above references proposed computational tools for estimating omnigenic interactome networks from a full set of SNPs in GWAS. Also, the tools have been used and validated in various practical applications.

ANSWER. Answer in pages 16-17 lines 442-446

Finally, the proposed QGSI framework directly addresses the growing recognition that epistasis and higher-order genetic interactions are fundamental components of complex trait architecture. Contemporary evidence from eco-evo-devo theory, omnigenic models, and multilayer interactome analyses shows that phenotypes emerge from dense networks of interacting loci whose effects are context-dependent and often nonlinear. Linear genomic selection indices can capture epistatic variance only indirectly through additive projections, whereas quadratic indices explicitly accommodate interaction-driven contributions via squared and cross-product terms. When combined with nonlinear genomic prediction methods—such as Gaussian kernel models that implicitly represent genome-wide epistasis—QGSI provides a biologically grounded and statistically coherent strategy for exploiting genetic networks without requiring explicit specification of interaction terms. This integration aligns with recent theoretical and empirical advances demonstrating that accounting for epistasis improves prediction accuracy, selection response, and long-term genetic gain in complex traits^{31,32,33,34}. Thus, QGSI should be viewed not merely as an extension of classical index theory, but as a network-aware genomic selection framework consistent with modern views of genetic architecture.

ANSWER. Page 27 lines 635-644

31. Flatt, T. & Heyland, A. Mechanistic insights into eco-evo-devo dynamics from genetic networks. *Nature Reviews Genetics* 23, 177–191 (2022). <https://doi.org/10.1038/s41576-021-00425-2>

32. Marbach, D. *et al.* Modeling genome-wide genotype-by-environment interactions through omnigenic interactome networks. *Cell Reports* 35, 109114 (2021). <https://doi.org/10.1016/j.celrep.2021.109114>

33. Mackay, T. F. C. Epistasis and quantitative traits: using model organisms to study gene–gene interactions. *Nature Reviews Genetics* 15, 22–33 (2014). <https://doi.org/10.1038/nrg3627>

34. Boyle, E. A., Li, Y. I. & Pritchard, J. K. An expanded view of complex traits: from polygenic to omnigenic. *Cell* 169, 1177–1186 (2017). <https://doi.org/10.1016/j.cell.2017.05.038>